# Blocking CCN2 Reduces Established Palmar Neuromuscular Fibrosis and Improves Function Following Repetitive Overuse Injury

**DOI:** 10.3390/ijms241813866

**Published:** 2023-09-08

**Authors:** Alex G. Lambi, Robert J. DeSante, Parth R. Patel, Brendan A. Hilliard, Steven N. Popoff, Mary F. Barbe

**Affiliations:** 1Department of Surgery, Plastic Surgery Section, New Mexico Veterans Administration Health Care System, Albuquerque, NM 87108, USA; alex.lambi2@va.gov; 2Division of Plastic Surgery, University of New Mexico School of Medicine, Albuquerque, NM 87106, USA; 3Aging + Cardiovascular Discovery Center, Lewis Katz School of Medicine at Temple University, Philadelphia, PA 19140, USA; robert.j.desante@gmail.com (R.J.D.); ppatel2015@temple.edu (P.R.P.); brendan.hilliard@temple.edu (B.A.H.); 4Department of Biomedical Education and Data Science, Lewis Katz School of Medicine at Temple University, Philadelphia, PA 19140, USA; steven.popoff@temple.edu

**Keywords:** CTGF, Pamrevlumab, FG-3019, muscle fibrosis, nerve fibrosis, enthesis, repetitive strain injury, grip strength, allodynia, mechanical hypersensitivity

## Abstract

The matricellular protein cell communication factor 2/connective tissue growth factor (CCN2/CTGF) is critical to development of neuromuscular fibrosis. Here, we tested whether anti-CCN2 antibody treatment will reduce established forepaw fibro-degenerative changes and improve function in a rat model of overuse injury. Adult female rats performed a high repetition high force (HRHF) task for 18 weeks. Tissues were collected from one subset after 18 wks (HRHF-Untreated). Two subsets were provided 6 wks of rest with concurrent treatment with anti-CCN2 (HRHF-Rest/anti-CCN2) or IgG (HRHF-Rest/IgG). Results were compared to IgG-treated Controls. Forepaw muscle fibrosis, neural fibrosis and entheseal damage were increased in HRHF-Untreated rats, compared to Controls, and changes were ameliorated in HRHF-Rest/anti-CCN2 rats. Anti-CCN2 treatment also reduced phosphorylated-β-catenin (pro-fibrotic protein) in muscles and distal bone/entheses complex, and increased CCN3 (anti-fibrotic) in the same tissues, compared to HRHF-Untreated rats. Grip strength declines and mechanical sensitivity observed in HRHF-Untreated improved with rest; grip strength improved further in HRHF-Rest/anti-CCN2. Grip strength declines correlated with muscle fibrosis, entheseal damage, extraneural fibrosis, and decreased nerve conduction velocity, while enhanced mechanical sensitivity (a pain-related behavior) correlated with extraneural fibrosis. These studies demonstrate that blocking CCN2 signaling reduces established forepaw neuromuscular fibrosis and entheseal damage, which improves forepaw function, following overuse injury.

## 1. Introduction

Upper extremity work-related musculoskeletal disorders, which are under the umbrella of overuse injuries, are a leading cause of long-term pain and disability [1,2,3,4]. These ultimately result in significant lost work time for patients and increased health care expenditures [5,6]. Temporary or permanent damage to soft tissues (nerve, muscles, tendons, and ligaments) can result from prolonged performance of work tasks with high-risk attributes, including high repetition, high force, and awkward postures [2,4,7]. Muscle, nerve, tendon, and even joint disorders can develop in humans following prolonged performance of intense repetitive tasks [4,7,8,9,10,11]. Effective treatments for established overuse-induced pathophysiological changes and ensuing loss of function are still needed.

We have developed a volitional rat model of overuse injuries in which rats perform a reaching and lever-bar pulling task for up to 24 weeks. The rats, motivated by obtaining a food reward, learn to pull the lever at defined reach rates and target forces [12]. Inflammatory changes begin early in a dose–response manner coincident with tissue microdamage, with more inflammatory responses and injury induced by high-repetition, high-force (HRHF) tasks than with less intense tasks [12]. Inflammatory responses observed during the development of overuse injuries include increased macrophages and pro-inflammatory cytokine production (e.g., IL-1α, IL-1β, and TNFα) in each tissue involved in performing this intense repetitive task (i.e., muscle, nerve, tendon, and bone) [12], as well as in distant tissues when the inflammatory response becomes systemic [13]. While these inflammatory responses may resolve (albeit cyclically), rats engaged in performing the HRHF tasks for 6 weeks or longer develop fibrotic changes in the fascia of forelimb tissues, such as in the median nerve (extra- and intraneural fibrosis), flexor muscles (intra- and peri-muscular fibrosis) and flexor tendons (epitendon fibroplasia and tendon disorganization) [14,15,16]. This increase in collagen deposition that is a hallmark of fibrosis was concomitant with increasing levels of transforming growth factor beta 1 (TGF-β1), alpha smooth muscle actin (αSMA), and cell communication network factor 2 (CCN2; also known as connective tissue growth factor (CTGF)) [16,17,18]. 

The finding of increased CCN2 in tissues showing enhanced collagen deposition was of interest since this protein has been shown to be a central mediator of tissue remodeling and fibrosis. For example, cell culture studies show that high levels of CCN2 increase the production and deposition of collagen subtypes by tenocytes, fibroblasts, and osteoblasts (all matrix-producing cells) [19,20,21,22]. In vivo, CCN2 immunoexpression is also seen in these same cell types and others (e.g., Schwann cells) [18,21,23,24]. CCN2 is a matricellular protein belonging to the six-member family of CCN2 proteins, grouped by their similar multi-modular structure. The CCN2 protein consists of four domains: (1) insulin-like growth factor 1 binding module; (2) von Willebrand factor type C repeat; (3) thrombospondin type 1 repeat; and (4) a cysteine knot motif. As a result of this complex structure, CCN proteins are able to exert various biological effects through interaction with other cells and extracellular matrix (ECM) components, including critical processes leading to fibrosis and skeletal tissue homeostasis [25].

A human monoclonal anti-CCN2 antibody, known as FG-3019 or Pamrevlumab, has been developed and is in phase 3 clinical trials for use in treating multiple diseases, including Duchene muscular dystrophy and idiopathic pulmonary fibrosis [26,27,28,29]. This antibody targets the second (von Willebrand factor type C) domain and in doing so inhibits cell proliferation and induces apoptosis in mesothelioma cells in vitro [30]. In vivo, the anti-CCN2 antibody abrogated skin fibrosis in a mouse model of systemic sclerosis, concomitant with reduced cellular expression of platelet-derived growth factor beta (PDFGR-β), αSMA, phosphorylated SMAD, and fibroblast specific growth factor 1 [31]. When used as a preventive treatment in our overuse injury model in weeks 2 and 3 of a 3-week HRHF task, systemic provision of the anti-CCN2 blocked the development of muscle and neural fibrosis in parallel with significant reductions in collagen in each tissue. Blocking CCN2 signaling also reduced muscle levels of CCN2, TGF-β1, and myofibroblasts that contribute to collagen production [15]. In bone, early provision of the monoclonal anti-CCN2 antibody enhanced osteoblast activity and reduced osteoclast activity, which, combined, prevented a loss in forelimb trabecular bone volume associated with three weeks of repetitive overloading [32]. These data show CCN2 is critical to the development of chronic soft tissue fibrosis and tissue degeneration.

However, as mentioned earlier, effective secondary therapeutic treatments are needed to reduce tissue fibrosis and degeneration that have already occurred. We have determined that rats performing the HRHF task for 18 weeks show significantly enhanced collagen deposition in all involved forelimb soft tissues [16,33,34]. In contrast, rats that had received anti-CCN2 treatment during a 6-week rest period that following this 18-week task show significant reductions in collagen types 1 and 3 in flexor muscles [34]. This was paralleled by reductions in CCN2, TGF-β1, and FGF2 levels, and reduced numbers of αSMA and phosphorylated ERK immunopositive cells, each elevated considerably in untreated 18-week HRHF rat muscles. We also observed increased muscle expression of another CCN family member, CCN3, a protein also known as NOV (nephroblastoma overexpressed) that has anti-fibrotic properties [35,36]. In nerves, the secondary treatment with anti-CCN2 reduced task-induced increases in CCN2, collagen type 1, and degraded myelin basic protein in the median nerve at the wrist [33]. Additionally, task-induced losses in forelimb trabecular bone volume were rescued in anti-CCN2-treated animals via enhanced osteoblast numbers and activity [37].

However, other than task-induced forepaw dermal fibrosis [16], we have yet to examine forepaw changes induced by prolonged performance of an intensive HRHF task, and the effectiveness of rest, with or without anti-CCN2, as a secondary intervention. This has high pre-clinical relevance since the integrity of forepaw/hand tissues is critical to upper extremity function. Thus, our first objective was to determine if 18 weeks of HRHF task performance induced fibro-degenerative changes in muscles, nerves, and entheses of the forepaw, and to determine whether rest, with or without concurrent anti-CCN2 treatment, would reverse or at least reduce any task-induced matrix changes. We hypothesized that collagen deposition and enthesis pathology would increase significantly in these tissues in untreated 18-week HRHF task rats, and that use of anti-CCN2 combined with rest as a therapeutic intervention would improve or restore forepaw tissue integrity. Our second objective was to determine if the forepaw fibro-degenerative responses correlated with changes in grip strength and forepaw mechanical sensitivity. Lastly, we continued to examine for changes in other fibrogenic and tissue repair contributors that might elucidate the mechanisms of action of this human anti-CCN2 monoclonal antibody in vivo.

## 2. Results

### 2.1. Task-Induced Increases in Collagen Desposition Are Present in Forepaw Muscle Connective Tissues; Deposition Is Reduced by Rest + Anti-CCN2 Treatment

After three weeks of performance of this same HRHF task, we have observed increased collagen deposition in rat forepaw muscular tissue (and associated connective tissues) involved in power gripping [33]. We extended that work here to first investigate the amount of collagen staining in forepaw muscles of 18-week HRHF rats (HRHF-Untreated), relative to rats that rested for 6 weeks after task cessation while simultaneously receiving either non-immune IgG injections (HRHF-Rest/IgG rats) or a monoclonal antibody directed against CCN2 (HRHF-Rest/anti-CCN2). Results were compared to age-matched controls that received IgG treatment (Control+IgG). The design of this study has been previously diagrammed [33,34,37].

The amount of collagen (blue after Masson’s trichome staining) was quantified using a thresholded pixel count method and the percent tissue area with collagen staining calculated. We observed increased collagen staining in forepaw muscles of HRHF-Untreated and HRHF-Rest/IgG rats, compared to Control rats (Kruskal–Wallis test, *p* = 0.004; post hoc results shown in Figure 1A). This was most prominent in muscle connective tissues (i.e., epimysial, perimysial and endomysial). The percent area with collagen staining in forepaw muscles of HRHF-Rest/anti-CCN2 rats was not statistically different from that of Controls (Figure 1A).

When the data were divided by region (thenar, hypothenar and deep instrinsics) and group, repeated-measures mixed-effect model analyses showed a significant difference between the region (*p* = 0.02), treatment groups (*p* < 0.0001), and their interaction (*p* = 0.02). Post hoc analyses revealed that the percent area with collagen staining was increased in thenar muscles of HRHF-Untreated rats, compared to Controls and HRHF-Rest/anti-CCN2 rats, and hypothenar muscles of HRHF-Untreated rats, compared to that of the other groups (Figure 1B). Deep intrinsic muscles (i.e., interossei) showed a similar but non-significant trend towards more collagen staining (Figure 1B).

Figure 1C–F shows representative examples of hypothenar muscles in the forepaw of each group. A small amount of blue staining, indicative of collagen deposition, was visible in connective tissues of Control+IgG hypothenar muscles; only loose areolar connective tissue was located between the individual muscles (Figure 1C). In contrast, there were regions with dense collagen between muscle fascicles, and between muscles and bone profiles in HRHF-Untreated rats (indicated by asterisks; muscles are lightly stained in the larger image panel so as to see the collagen more clearly; Figure 1D and inset). The HRHF-Rest/IgG rats that had ceased performing the task and that had rested for 6 weeks with concomitant IgG treatment continued to show enhanced collagen deposition around muscle fascicles, with some forepaws showing considerable endomysial collagen deposition, i.e., between individual myofibers (Figure 1E). The HRHF-Rest/anti-CCN2 muscles resembled those of Control rats and had less collagen staining overall, less around individual myofibers, and less between muscles and bone than seen in HRHF-Untreated palms (Figure 1F).

Figure 1G,H show representative examples of collagen type I immunostaining (green fluorescence) in the connective tissue regions between muscles and bone of Control+IgG and HRHF-Untreated rats. Note the dense collagen fibrils in this region in HRHF-Untreated rats that are missing from the Control+IgG rats. This matches previously reported data of increased collagen type I assayed using ELISA in muscles of HRHF-Untreated rats, compared to Control+ IgG and HRHF-Rest/anti-CCN2 rats (See Appendix A and ref. [34]). Appendix A also shows that collagen type 1 contributes more to the overall amount of collagen in these muscles than collagen type 3.

Thus, the established collagen deposition (primarily collagen type 1) in forepaw muscles, induced by prolonged task performance, was ameliorated by the 6 weeks of rest with anti-CCN2 treatment.

### 2.2. Task-Induced Increases in Fibrosis-Related Proteins in Muscles Are Reduced by Rest + Anti-CCN2 Treatment

In an effort to explore the underlying mechanisms in our model, we have examined expression levels of several pro-fibrogenic proteins in flexor muscles using ELISA and Western blot methods. We have reported that levels of CCN2, FGF-2, and TGF-β1 are significantly increased in HRHF-Untreated and HRHF-Rest/IgG rat muscles, compared to those of Control rats, yet are at control levels in HRHF-Rest/anti-CCN2 rat muscles (See Appendix A and reference [34]).

We extended that work here to examine protein expression levels of phosphorylated β-catenin, the activated version of β-catenin, which is another pro-fibrogenic protein [38,39]. Using Western blot methodology, a specific antibody to phosphorylated β-catenin detected ~92 and ~86 kDa bands [40,41] (Figure 2A). Densitometry was used to analyze the ~92 and ~86 kDa bands. A repeated measures mixed model was used to analyze the data, and showed a significant difference between the ~92 and ~86 kDa bands (*p* = 0.01) across the groups. Post hoc analyses revealed that the ~92 kDa band was increased in HRHF-Untreated muscles, compared to Control muscles (lane 5). This same band was lower in HRHF-Rest/anti-CCN2 muscles, compared to HRHF-Untreated and HRHF-Rest/IgG muscles, as shown in Figure 2A,B,D. However, ~86 kDa band was not altered across the groups (Figure 2A,B,D), nor was nonactive β-catenin (Figure 2C). 

We also examined levels of CCN1, another CCN family member with pro-fibrogenic properties [42,43,44] although it is known to be only transiently produced after mechanical loading [45,46]. A band at ~38 kDa was observed in muscles of each group. There was a trend towards an increase in HRHF-Untreated muscles compared to those of Controls (*p* = 0.056), although the overall ANOVA was not significant (*p* = 0.19). These results are shown in Appendix A.

Combined with our past results [34], the combination of rest and anti-CCN2 treatment significantly lowered muscle levels of several pro-fibrogenic proteins, including TGF-β1, FGF-2, and phosphorylated β-catenin, as well as CCN2 and perhaps the partial drop in CCN1, each of which were elevated in HRHF-Untreated rat muscles.

### 2.3. Task-Induced Increases in Myofibroblast Numbers Are Decreased by Anti-CCN2 Treatment

TGF-β1 and phospho-β-catenin increases, both observed in muscles in our model (see Appendix A and Figure 2 above) are often associated with increased myofibroblasts, often detected as αSMA immunopositive (+) cells [16,47]. Therefore, we examined the muscles for αSMA+ fibroblastic cells. As seen in representative images of forelimb flexor muscles in cross-section (Figure 3 (left column)), many αSMA+ cells were located on the perimeter of myofibers in HRHF-Untreated muscles, relative to Control muscles (Figure 3B vs. Figure 3A). Blocking CCN2 signaling was associated with fewer αSMA+ cells in the HRHF-Rest/anti-CCN2 muscles (Figure 3C). We also observed increased numbers of αSMA+ cells in the connective tissues external to the muscles and tendons of HRHF-Untreated rats (Figure 3B (right column)), cells not present in Control or HRHF-Rest/anti-CCN2 rats (Figure 3A,C (right column)). Quantification of these myofibroblast-like cells confirms these observations and shows that the HRHF-Rest/IgG results resembled those of HRHF-Untreated animals (Table 1). Thus, anti-CCN2 combined with rest lowered numbers of myofibroblastic-like cells in muscle and related connective tissues that had been elevated by prolonged task performance.

### 2.4. CCN3, Which Has Anti-Fibrosis Properties, Increased in Muscles with Anti-CCN2 Treatment

We extended our studies, examining the effects of task and treatment on CCN3 on muscle, another member of the CCN family with anti-fibrotic properties [43,48,49]. We first verified the anti-CCN3 antibody with a Western blot loaded with recombinant CCN3 (rCCN3) protein (Figure 4A (left)). Bands at ~36 and ~28 kDa were detected. Pre-absorbing the CCN3 antibody with rCCN3 protein blocked their detection (but not a ~50 kDa band that we did not assess further). This antibody was then used to probe Western blots of muscle lysates. Three bands were detected at ~39, ~36, and ~28 kDa (matching bands at molecular weights identified by others as truncated forms of CCN3 [50,51,52]). Their highest expression was in HRHF-Rest/anti-CCN2 and Control+IgG muscles (Figure 4B). Densitometry of replicate blots followed by repeated-measures mixed-model analyses (with bands and group as factors) revealed a significant group effect (*p* = 0.03). Post hoc analysis showed increases in the ~39 band in HRHF-Rest/anti-CCN2 muscles, relative to HRHF-Rest/IgG, and the ~36 kDa band in HRHF-Rest/anti-CCN2 muscles, relative to HRHF-Untreated and HRHF-Rest/IgG (Figure 4D, each *p* < 0.05). The ~36 and ~28 kDa bands were decreased in HRHF-Untreated and HRHF-Rest/IgG, compared to Control muscles (Figure 4D, each *p* < 0.05).

### 2.5. Matrix Metalloproteinases (MMPs) 1, 2, and 9 Are Not Affected by the Interventions

It has been suggested that CCN family members and β-catenin interact with various MMPs [38,49]. Therefore, we checked for levels of MMP-1, -2, and -9. Gelatin zymograms showed no active MMP-9 (which should be at ~92 kDa) in the muscle lysates of any group (Figure 5A). While both pro- and active MMP-2 (~72 and ~63 kDa, respectively) are present in the muscles, no differences were seen between the groups (Figure 5A). Western blot methodology was used to examine for MMP-1 protein expression (Figure 5B). Each task and treatment group appeared similar, each with some decline in MMP-1 relative to Control muscles. Replicates of these gels showed similar results. These findings rule out MMP-1, -2, and -9 as contributors to the reduced collagen deposition observed in the HRHF-Rest/anti-CCN2 muscles.

### 2.6. Extraneural Fibrosis after Task Performance Is Decreased by Rest + Anti-CCN2 Treatment

During the forepaw muscle examinations, we also observed increased collagen staining around nerve branches in the forepaw region, matching past findings of increased extraneural fibrosis more proximally in forelimb tissues, with prolonged performance of intense tasks in this model [32]. This was quantified in the palmar region and showed an increase in percent area with collagen staining around nerve branches of HRHF-Untreated rats, compared to those of Control and HRHF-Rest/anti-CCN2 rats (Kruskal–Wallis test, *p* < 0.0001, with Dunn’s post hoc results shown in Figure 6A). While rest alone (HRHF-Rest/IgG) resulted in a decrease in collagen staining, this reduction was not significant. Representative images from each group are shown in Figure 6B–F. Very little extraneural collagen staining was visible in Control rats (Figure 6B). In contrast, the HRHF-Untreated and HRHF-Rest/IgG showed increased collagen staining around branches of both median and ulnar nerves in the forepaw (i.e., extraneural; Figure 6C–E). This collagen staining was reduced in HRHF-Rest/anti-CCN2 nerves (Figure 6F). Immunohistochemistry for collagen type 1 showed that this key collagen subtype increased in nerves of HRHF-Untreated rats relative to Control rat nerves (Figure 6G,H), matching past results for our model [18,33].

Thus, the established collagen deposition (primarily collagen type 1) around forepaw nerves, induced by prolonged task performance, was ameliorated by the rest with anti-CCN2 treatment.

### 2.7. Task-Induced Degenerative Entheseal Changes Are Decreased with Rest + Anti-CCN2 Treatment

Since we have previously reported task-induced degeneration of distal forelimb trabecular bone in HRHF-Untreated rats that was ameliorated by the anti-CCN2 treatment [34], we next examined for degenerative changes in entheses of the forepaw and wrist where muscle/tendon units insert and ligaments attach. We divided these into laterally and medially located entheses: (1) lateral forepaw or wrist entheses (i.e., on distal radius and/or laterally located carpal and metacarpal bones), or on (2) medial forepaw or wrist entheses (i.e., on distal ulna and/or medially located carpal and metacarpal bones). These entheses were graded binarily (1 indicating present) on six domains: tidemark changes, underlying bone remodeling, fissuring, void space, vascular invasion, attachment site holes. In laterally located entheses, significant group effects were observed by treatment group (*p* < 0.0001), enthesis feature (*p* < 0.0001), or their interaction (*p* = 0.004). Main effects post hoc analyses revealed that the overall enthesis score of damage was significantly increased in HRHF-Untreated rats, compared to that in Control and HRHF-Rest/anti-CCN2 rats (Figure 7A). Tukey’s multiple comparison post hoc analyses revealed (1) the presence of fissures was higher in HRHF-Untreated rats, compared to that in Control and HRHF-Rest/anti-CCN2 rats (*p* = 0.002 and *p* = 0.007, respectively); (2) tidemark changes were greater in HRHF-Untreated rats compared to those in Controls (*p* = 0.004); (3) underlying bone remodeling was greater in HRHF-Untreated rats compared to that in Control and HRHF-Rest/anti-CCN2 rats (*p* = 0.0002 each); and (4) the presence of holes at the attachment site was greater in HRHF-Untreated rats compared to that in Controls (*p* < 0.0001). In medially located entheses, while the changes were less severe than in laterally located entheses (Figure 7B), significant effects were observed by treatment group (*p* = 0.008) and enthesis feature (*p* = 0.0003), but not their interaction (*p* = 0.33). Main effects post hoc analyses revealed that the overall enthesis score of damage in medial forepaw/wrist locations was increased in HRHF-Untreated rats compared to that in Control and HRHF-Rest/anti-CCN2 rats (Figure 7B). Further post hoc analyses revealed (1) tidemark changes were slightly higher in HRHF-Rest/IgG rats compared to those in Controls (*p* = 0.049); and (2) underlying bone remodeling was greater in HRHF-Untreated rats compared to that in Controls (*p* = 0.02).

Thus, several entheseal degenerative changes were observed in HRHF-Untreated rats that were ameliorated by the anti-CCN2 treatment combined with rest.

### 2.8. Changes in Phosphorylated β-Catenin and CCN3 May Play a Role in the Entheseal Improvements with Anti-CCN2 Treatment

We next examined if changes in β-catenin or CCN3 levels may have contributed to the entheseal responses, since the former has been implicated in muscle–bone cross-talk and the latter in skeletal tissue repair [53,54]. Western blots were used to examine distal forelimb and carpal bones to which the entheses were attached (termed bone + entheses in Figure 8). We saw that nonactive β-catenin (~92 kDa) did not alter between the HRHF groups, with or without intervention (Figure 8A (left)). However, a specific antibody to phosphorylated β-catenin detected a ~92 kDa band that was highest in HRHF-Untreated bone + entheses (Figure 8A). Densitometry revealed that the ~92 kDa band of phosphorylated β-catenin bands was increased in HRHF-Untreated bone + entheses (*p* < 0.05), compared to both HRHF-Rest treated groups. For further confirmation of band specificity, we loaded lysates from primary fibroblasts into lanes 10–12 (primary fibroblasts were grown in media with serum as previously described for primary tenocytes [19]). Only a ~92 kDa band of β-catenin antibody was detected.

We also examined the effects of task and treatment on CCN3 on the same bone and entheses lysates. Two bands were detected at ~39 and ~36 kDa in the rat bone + entheses of HRHF rats, with or without interventions, yet none were detected in the Control lysates (Figure 8D). Densitometry revealed that the ~36 kDa band of CCN3 increased above Control levels in the HRHF-Rest/CCN2 rat bone + entheses (*p* < 0.05; Figure 8D–F).

### 2.9. Functional Changes Correlate with Histopathological Findings of Forepaw Neurodegenerative Changes

Since functional declines, pain, and discomfort are seen in patients with chronic overuse injuries [8], we next examined the rats’ behavioral data collected at baseline and prior to euthanasia and tissue collection. We analyzed for percent change in grip strength from baseline levels, and found a significant decrease in HRHF-Untreated compared to Control and HRHF-Rest/anti-CCN2 groups (Kruskal–Wallis test *p* = 0.003; Dunn’s post hoc results shown in Figure 9A). The magnitude of percent declines in grip strength (a decrease in grip strength from baseline levels) correlated with the quantity of fibro-degenerative changes: muscle fibrosis in thenar or hypothenar muscles (Figure 9B, r = −0.41), overall enthesis scores (Figure 9C, r = −0.40), and forepaw nerve fibrosis (Figure 9D, r = −0.51). We have also reported a significant decrease in nerve conduction velocity (NCV) in the median nerves at the wrist in the HRHF-Untreated rats, compared to that in Controls, that was rescued by the anti-CCN2 treatment [33]. Therefore, we also performed a Pearson’s r correlational analysis of percent change in grip strength versus these same rats’ NCV results. We found that the percent change in grip strength correlated strongly with NCV (r = 0.62; specifically, as NCV declined, grip strength declined, and as NCV increased, grip strength increased; Figure 9E).

Lastly, we examined the percent change in the rats’ forelimb withdrawal response to 1 g (0.98 cN) and 4 g (3.92 cN) probing of the mid-palmar region, used as a forepaw mechanical sensitivity test (i.e., forepaw hypersensitivity to a non-noxious touch stimulus). There was an increase in percent change in forepaw mechanical sensitivity to a 1 g monofilament in HRHF-Untreated rats compared to that in other groups (Kruskal–Wallis test *p* = 0.002; Dunn’s post hoc results shown in Figure 10A), and to a 4 g monofilament in HRHF-Untreated rats compared to that in HRHF-Rest/IgG and HRHF-Rest/anti-CCN2 rats (Kruskal–Wallis test *p* = 0.003; Dunn’s post hoc results shown in Figure 10B). We observed a moderate correlation between the response to a 1 g monofilament and % collagen around forepaw nerve branches (Figure 10C, r = 0.55), although a weak correlation between the response to a 4 g monofilament and % collagen around forepaw nerve branches (Figure 10D, r = 0.32). These results indicate increased forepaw hypersensitivity to a 1 g monofilament with increasing collagen deposition around palmar nerves.

Thus, these data combined indicate that enhanced collagen deposition in extracellular matrices as well as entheseal degenerative changes negatively affect forearm function in untreated task rats. Reductions in these fibro-degenerative tissue changes with rest combined with the anti-CCN2 significantly improved overall forearm function.

## 3. Discussion

This is the fourth study of a series examining tissues from the same animals [33,34,37]. Unique to this study is the examination of (1) collagen deposition in forepaw muscles and nerves; (2) histopathological changes in entheses through which involved muscles attach to underlying bones; (3) sensorimotor changes (forepaw mechanical allodynia and forelimb reflexive grip strength) reported as percent change from baseline levels by study end (24–30 weeks later), and whether these behavioral changes correlate with forepaw tissue histopathology. Additionally, we examined for levels of (4) the pro-fibrogenic protein, β-catenin (nonactive and phosphorylated) in flexor muscles and in distal forelimb and carpal bones to which the entheses were attached; (5) MMP-1, -2, and -9 in flexor muscles; and (6) a protein with skeletal tissue repair properties, CCN3, in the distal forelimb bone + entheses. We present additional data on levels of CCN1 and CCN3, and numbers of myofibroblastic cells in the flexor muscles involved. In vivo nerve conduction velocity testing was performed in the last week before euthanasia in a prior study [33]. Those data were reused in statistical correlations with grip strength data.

Power gripping involves many forepaw tissues. Thus, we sought to determine for the first time if 18 weeks of HRHF task performance induced fibrotic changes in muscles in the forepaw, in nerves located in the forepaw that innervate these forepaw tissues and skin, and in entheses where these muscles attach to underlying bones, and also if these potential changes could be reversed by 6 weeks of rest combined with anti-CCN2 treatment. We observed increased forepaw muscle and nerve fibrosis and entheseal damage in the HRHF-Untreated animals. These fibro-degenerative changes were ameliorated in the Rest/anti-CCN2 treated rats. This treatment also reduced protein expression levels of phosphorylated-β-catenin (a Wnt pathway protein shown to contribute to the pathogenesis of fibrotic disorders [55]) in muscles and the distal bone/entheses complex, increased expression of CCN3 (a CCN protein family member with tissue healing properties, including reductions in fibrosis [48,49]) in the same tissues, and reduced αSMA+ cells in flexor muscles, compared to HRHF-Untreated rats. In contrast, the Rest/IgG treatment reduced nerve fibrosis, but not muscle fibrosis (although it did reduce αSMA+ cells), and only partially reduced entheseal damage. Grip strength declines and mechanical sensitivity observed in HRHF-Untreated improved with rest alone, while grip strength improved even further with Rest/anti-CCN2 treatment. Thus, blocking CCN2 signaling reduced established forepaw neuromuscular fibrosis and entheseal damage, and improved forepaw function.

Repetitive strain injuries result from biomechanical stress and tissue microtrauma [56,57]. Mechanical stress through collagen fibrils and bundles induces physiological responses in matrix cells (e.g., fibroblasts, tenocytes and osteoblasts) via both mechanotransduction and modulation of gene expression patterns [20]. We have reported increased collagen deposition in forepaw muscular tissue (including associated connective tissues, which includes the epi-, peri-, and endomysium) involved in power gripping in rats performing this same HRHF task for only three weeks [33]. The amount of muscle fibrosis was considerably lower at 3 weeks than that seen in these 18-week HRHF-Untreated rats. In addition, no entheseal changes were detected in a study examining entheseal changes in 3-week HRHF-Untreated rats [32]. This is in sharp contrast to the entheseal microdamage observed in the entheses of the 18-week HRHF-Untreated rats of this study. This makes sense since entheses are one of the fundamental elements of musculoskeletal tissue through which muscles transmit their forces to underlying bones [58]. Enthesopathies are considered as musculoskeletal stress markers and are assumed to reflect the activity of the attaching muscles even in anthropological studies [59,60]. Humans engaged in heavy manual tasks also show more entheseal lesions than individuals engaged in nonmanual or light manual work [61]. 

The Rest/IgG treatment reduced nerve fibrosis, but not muscle fibrosis (although it did reduce αSMA+ cells) and only partially reduced the entheseal damage. Additionally, the Rest/IgG treatment did not reduce task-induced increases in phosphorylated-β-catenin in the muscles and did not increase CCN3 levels that had been dampened by task performance. Moreover, Rest/IgG treatment improved the mechanical sensitivity to monofilament forepaw skin probing, but only partially improved grip strength. In several other studies, we have shown that rest alone after HRHF-induced injury was an insufficient treatment choice for recovery of tissue integrity or function [14]. Rest alone was also shown to be a poor treatment for muscle fibrosis in another rat model of repetitive strain injury [62].

We have previously reported that after 18 weeks of HRHF task performance, utilization of the Rest/anti-CCN2 treatment (FG-3019, administered peritoneally) reduced collagen deposition around muscles and median nerve branches at the wrist that had been induced by 18 weeks of HRHF task performance [33,34]. It also reduced task-induced elevation of forelimb flexor muscle levels of Collagen types 1 and 2, FGF-2, TGF-β1, and itself (i.e., CCN2 levels), as well as increased quantities of αSMA and phosphorylated ERK in muscles [33,34]. Use of FG-3019 in other animal models of tissue fibrosis also showed its ability to reduce tissue fibrosis and improve function. In a murine model of amyotrophic lateral sclerosis (hSOD1^G93A^ mice), FG-3019 improved skeletal muscle structure (reduced fibrosis and neuromuscular junction cell–cell communication appeared to be enhanced) and locomotor function, when provided to these mice for 2 months, 3 times weekly, using intraperitoneal injections, beginning when they were 8 weeks of age [63]. In a preclinical mouse model of systemic sclerosis, FG-3019 treatment reduced skin fibrosis and cellular expression of a number of fibrogenic related proteins, including platelet-derived growth factor receptor beta (PDGF-β), procollagen, αSMA, phosphorylated Smad2, and fibroblast specific growth factor 1 when provided 3 times weekly using intraperitoneal injections, for two weeks, to CCN2/CTGF knock out mice [31]. In a mouse model of Duchenne muscular dystrophy (Mdx mice), FG-3019 reversed muscle fibrosis and restored skeletal muscle function [64]. Mdx mice with hemizygous CCN2 deletion, or treatment with a monoclonal antibody that targets the von Willebrand Factor type C (vWC) domain of CCN2, also show reduced muscle fibrosis and improved locomotion and muscle strength [64,65]. Each of these primary interventions occurred with few side effects.

The CCN family members have roles in multiple tissue types throughout the body, interact with and modulate multiple proteins, and are considered as some as the principal coordinators of extracellular matrix remodeling [49,66]. CCN2 directly interacts with CCN3 and TGF-β, for example [66,67]. CCN2 is known to have autocrine feedback loops on itself and its binding partners [25,66,67], so that inhibition of its function results in a decrease in the expression of CCN2, but also in the many partners with whom it interacts. The changes in phosphorylated β-catenin and CCN3 observed in this study may be the result of downregulation of CCN2 after the anti-CCN2 (FG-3019) treatment [34] (discussed further below), or unknown physical interactions between CCN2 and 3 that mediate their antagonistic effects [49,68,69].The anti-CCN2 (FG-3019) could also act as a clearance antibody, removing CCN2 from the vicinity of the cells or interfering in its interactions with some of its putative binding partners [70].

One of these binding partners is β-catenin. Collagen deposition in fibrotic tissues is known to result from activation of the Wnt canonical pathway, a pathway whose main function is carried out by β-catenin [71]. Sustained increases in β-catenin levels are linked to the pathogenesis of fibrosis, including in lung, kidney, uterine endometriosis, skin, and liver [38,72] by enhancing fibroblast proliferation and migration, and TGF-β, CCN2, and collagen production [73]. Important to our results is that the Wnt/β-catenin pathway works in a collaborative manner with TGF-β and CCN2 pathways in the process of fibrosis [38,55], although the cross-talk between the Wnt/β-catenin and TGF-β pathways is complex and context dependent [55,74]. CCN2 could even be the target gene of Wnt signaling by triggering the activation of TGF-β and CCN2 in fibroblasts and other cells in the fibrotic environment [55]. We further suggest that similar to TGF-β, CCN2 can promote the expression of Wnt/β-catenin family members [38], so that when CCN2 levels decreased with anti-CCN2 treatment, so did levels of phosphorylated β-catenin. This decrease in phosphorylated β-catenin may have contributed to muscle improvement, since a genetic study in adult satellite cells demonstrated that the silencing of Wnt/β-catenin signaling is important to muscle regeneration [75]. Wnt signaling also plays a role in bone homeostasis with osteoblast-derived Wnt enhancing osteoclast differentiation [53]. Its role in the enthesis needs further exploration.

CCN3, also known as NOV (nephroblastoma overexpressed), is a sister molecule with antagonist effects from CCN2. It has anti-fibrotic and anti-proliferative properties [36,49], is implicated in tissue repair [54], and is a negative regulator of CCN2 in a diabetic nephropathy model and mouse scleroderma model [48,76]. CCN3 is able to upregulate MMP1 and 3 proteases that contribute to matrix degradation (although we did not observe an increase in MMP1 in the muscles) [54,77]. These attributes support the hypothesis that the loss (or in our case, reduction) in CCN3 expression is permissive for fibrosis [36]. There also appears to be a mechanical component to CCN3 signaling, so that in mechanically strained fibroblasts, CCN3 levels are low during bouts of mechanical stress and increase again after relaxation [78]. The latter finding matches data shown here that CCN3 (three bands at ~39, ~36, and ~28 kDa) is reduced in muscles of HRHF-Untreated and HRHF-Rest/IgG treated animals, compared to in Control animals, and recovered to Control levels in HRHF-Rest/anti-CCN2 treated muscles. These truncated forms of CCN3 are thought to be biologically active [51,79]. While several bands were detected in muscles, only the ~36 kDa band was increased in the bone/entheses. These data suggest that there are isoform differences between tissues. The increase in CCN3 in entheses of HRHF-Rest/anti-CCN2 rats may contribute to the restored integrity of the entheseal tissues since CCN3 plays a significant role in maintaining articular cartilage, and osteoarthritic articular cartilage shows low levels of CCN3 [67]. Administration of recombinant CCN3 even ameliorates matrix catabolism by MMP1 [80]. Similar to β-catenin, CCN3’s role in entheses needs further exploration.

Regarding muscle function, we found that grip strength declines correlated with forepaw muscle and extraneural fibrosis, entheseal damage, and nerve conduction velocity. This was similar to a past study using this model in which decreasing grip strength correlated strongly with increasing muscle levels of procollagen 1 (r = −0.88), TGF-β1 (r = −0.83), and CCN2 (r = −0.55) [81]. It has been shown in another repetitive strain injury animal model that fibrosis can affect muscle function [57,62,82]. This has also been suggested for humans [58]. Grip strength declines also occur with enhanced inflammatory mediators in neuromuscular tissues [83,84]. Yet, pro-inflammatory cytokine levels in muscles are resolved in 18-week HRHF rats [13,16], thus ruling out muscle-derived inflammatory cytokines as contributors to the observed grip strength declines in this study.

Several studies have implicated nerve fibrogenic changes as contributors to functional declines [17,57,81,82,85]. It is logical that decreased conduction velocity of the median nerve can contribute to grip strength declines. One potential cause of nerve conduction velocity declines is extraneural fibrosis [85]. Monkeys performing a voluntary forceful repetitive pinching task develop declines in nerve conduction with continued task performance that corresponds with MRI-detected enlargement of median nerves near the proximal end of the carpal tunnel [86]. MRI detected enlargements of nerves can be due to either inflammation or extraneural fibrosis. Whichever the cause, cessation of the task was a successful treatment for recovery of nerve conduction in that study [86]. Six weeks of rest with IgG did not lead to recovery of nerve conduction in this study, and only partial improvement in grip strength [33]. Yet, Rest/anti-CCN2 treatment restored grip strength. We hypothesize that the greater functional recovery was due to a greater reduction in a mechanical constraint of neuromuscular tissues by thickened fascial tissues. While to date there have been no human trials specifically studying the role of anti-CCN2 therapy on extraneural fibrosis, a comparison of restrictive soft tissue effects can be made with studies in pulmonary fibrosis where anti-CCN2 therapy attenuates functional declines by decreasing interstitial lung fibrosis [28]. In a single-arm trial in non-ambulatory patients with Duchenne muscular dystrophy, patients treated with anti-CCN2 therapy had lower biceps brachii muscle fibrosis scores and improved upper limb function and grip strength compared to historic controls [29]. These findings, combined with the earlier discussed improved locomotor function and muscle strength following anti-CCN2 therapy in a murine model of amyotrophic lateral sclerosis [63] and Mdx mouse studies [64,65] in conjunction with reduced muscle fibrosis, implicate soft tissue fibrosis and degeneration as key contributors to functional declines, and anti-CCN2 therapy as a means to improve tissue function in these instances.

Median neuropathies can also contribute to enhanced pain behaviors (e.g., the enhanced mechanical allodynia seen in this study), as can spinal cord central sensitization changes (also shown in our model [13,33]). We suggest that prolonged task performance creates a chronic constriction of the nerve as a result of the enhanced collagen deposition in tissues that should be loose areolar connective tissues, similar to an experimental chronic nerve constriction model [85]. Thus, any reduction in collagen in fascial tissues surrounding the nerves should reduce pain behaviors, such as those occurring after rest, with or without anti-CCN2 treatment as shown in Figure 10. The Rest/anti-CCN2 treatment also reduces sensitization of dorsal root ganglion neurons and the expression of a neuronal stress marker (activating transcription factor 3, ATF3) in motor neurons located in the spinal cord more than rest alone [33]. Improved neural functioning was also observed in the murine model of amyotrophic lateral sclerosis after CCN2 treatment, as evidenced by enhanced neuromuscular junction cell–cell communication [63].

To date, antibodies to CCN2 are administered systemically for fibrotic disorders of a number of types in humans and *in vivo* animal models [15,27,32,33,34,37,87,88,89,90]. This is an advantage for the treatment of overuse injuries that can effect a variety of different tissues (tendon, nerve, muscle, and bone) and multiple anatomical sites dependent on the work task [4,7,10,11,91,92,93], sometimes simultaneously in the same patient [8,91]. In our rat model of overuse injury, repeated and persistent inflammation and subsequent fibrogenic processes were found in forepaw and forelimb nerves, muscles, tendons, and associated connective tissues [13,15,16], while degenerative changes have been detected in tendon and bone [32,37]. We postulate that the systemic administration of a drug is key to recovery of function in these types of disorders.

Limitations to this study include the use of female rats only. Although the female rats do not show significant changes in serum estradiol levels across the weeks of task performance or with Rest/IgG or Rest/anti-CCN2 treatment [94], caution should still be taken regarding how these results can be extrapolated to male rats. Additionally, this is a rat study. Although an anti-fibrosis treatment is greatly needed for humans with work-related or other musculoskeletal disorders [95], the anti-CCN2 drug used here has yet to be tested in humans for overuse-induced musculoskeletal disorders. 

## 4. Materials and Methods

### 4.1. Animals

The animal study protocol was approved by the Institutional Animal Care and Use Committee in compliance with National Institutes of Health guidelines. As previously described and diagrammed in design figures [33,34,37], studies were conducted on 31 young adult, female, Sprague–Dawley rats (3 month of age at onset and 7.5 to 9 months of age at completion; Charles River Associates, Wilmington, MA, USA).

Rats were randomly divided into one of four groups:

(1) Age- and weight-matched Controls (*n* = 10);

(2) Task rats that learned and then performed an HRHF task for 18 weeks (HRHF-Untreated, *n* = 10);

(3) and (4) Task rats that performed the HRHF task for 18 weeks before cessation of the task for 6 weeks with simultaneous systemic treatment with an anti-CCN2 antibody (HRHF-Rest/anti-CCN2, *n* = 6), or with a non-immune human immunoglobulin (HRHF-Rest/IgG, *n* = 6) before euthanasia. One of the HRHF-Rest/IgG group died due to unknown reasons in week 12, thereby reducing the number of animals in this latter group to *n* = 5.

Food restriction less than 5% of baseline was needed to motivate the animals to pull for a food reward, as previously described [32]. Yet, all rats were allowed to gain weight over time, as previously reported [37]. Sentinel rats showed no indices of illness across the weeks of task performance.

### 4.2. Operant Training and HRHF Task Performance

The details of the HRHF task apparatus, initial learning period, and then operant task performance of this repetitive reaching task have been previously reported [12,37]. Briefly, task rats learned a reaching and lever-pulling task at high force loads at no specified reach rate (ramping upwards from naïve, 10 min/d, and 5 d/wk), across a 6-week learning period. They then went on to perform a reaching and lever-pulling task for 18 weeks for a food reward: they pulled on the lever bar at 48% of their maximum pulling force (1.21 N), at 4 reaches/min, 2 h/d, in four 30-min intervals per task day (with a 1.5 h break between sessions), 3 d/wk, and for 18 weeks. The animals were ambidextrous from the onset and utilized either forelimb to pull the lever bar interchangeably across each session, day, and week. 

### 4.3. Pharmacological Treatments

The specifics of the pharmacological treatments were as previously described [33]. Briefly, HRHF-Rest/anti-CCN2 rats performed the HRHF task for 18 weeks, before resting for 6 weeks while being treated twice/wk with a human anti-CCN2 monoclonal antibody (FG-3019, FibroGen, Inc., San Francisco, CA, USA; 40 mg/kg body weight final dose via intraperitoneal injections (i.p.)), with the dose and route of administration according to manufacturer’s suggestions for rats. The HRHF-Rest/IgG rats also performed the HRHF task for 18 weeks, before then resting for 6 weeks while being treated twice/wk for 6 wks with a non-immune human immunoglobulin (IgG; FibroGen; same amount of liquid as for FG-3019, i.p.) as a vehicle control. The control animals were similarly treated with the IgG. No adverse effects of the treatment were observed, as previously reported [15,32,33,34,37].

### 4.4. Behavioral Assays

All rats in the study underwent behavioral assays for grip strength and forepaw mechanical sensitivity, bilaterally. Reflexive grip strength was tested using a rat grip strength meter (1027SR-D58, Columbus Instruments, Columbus, OH, USA) after onset of food restriction (to serve as the baseline measurement), and at the endpoint within three days of euthanasia and tissue collection (week 18 for the HRHF rats, and week 24 for the HRHF-Rest/anti-CCN2 and HRHF-Rest/IgG rats). The test was repeated five times/limb during each testing session. The percent change in grip strength from baseline levels was calculated by subtracting the final results number from the baseline results, dividing that number with the baseline result, and then multiplying by 100. Mechanical sensitivity of the forepaw in response to 1 g (0.98 cN) and 4 g (3.92 cN) monofilament probes was performed as previously reported [96] at the same timepoints as the grip strength assays. All rats pulled on the lever bar with both hands, that is, all animals were ambidextrous. Therefore, behavioral data from each limb were separately included in the statistics.

### 4.5. Tissue Collection

Animals were deeply anesthetized with 5% isoflurane in oxygen, and then euthanized by performing thoracotomy and cardiac puncture for blood collection using a 23-gauge needle. Serum was harvested from blood, as described, for other studies [32,33,34]. In one forearm, muscles, the distal radius and ulna (metaphyseal and epiphyseal regions), and first row of carpal bones with attached entheses were first removed. These were flash frozen and used for the biochemical assays described further below. Thereafter, the rats were transcardially perfused with buffered 4% paraformaldehyde, as described [34]. In that same limb, the remainder of the forepaw (2nd row of carpal bones and distally, with attached entheses and muscles) was immersion fixed in buffered 4% paraformaldehyde before being washed in water and decalcified (NC9044643, StatLab Immunocal solution, Fisher Scientific, Waltham, MA, USA). These forepaws were processed for paraffin embedding, as previously described [33]. In the contralateral intact forearm, forepaws and attached distal forelimb tissues were collected and underwent plastic embedding.

Thus, tissues were collected bilaterally from each rat of each group: 10 age-matched Control, 10 HRHF-Untreated, 6 HRHF-Rest/anti-CCN2, and 5 HRHF-Rest/IgG rats.

### 4.6. Neuro-Muscular Histomorphometry of Trichrome Stained Collagen Deposition

Forepaws and forelimb prepared for plastic embedding were processed and embedded in methyl methacrylate resin (MMA, Osteo-Bed Bone Embedding Kit, Polysciences, Warrington, PA, USA, or by Bioquant Inc., Nashville, NC, USA), as previously described [32,94]. Paraffin-embedded forepaws were sectioned into 5 um longitudinal sections and placed on charged slides (Tissue Path Superfrost Slides, Fisher Scientific). Enough sections on slides were prepared such that there were at least 3 replicates for each animal and limb side.

Sections on slides were deplasticized for 3 days in xylene at 37 °C or deparaffinized overnight in xylene at room temperature. Sections of forepaws collected from all rats in the study underwent trichrome staining (Masson’s or Goldner’s). The individuals carrying out all histomorphometric analyses were blinded to treatment. At least 3 different areas of muscle/nerve were scored per animal. This was performed using a Nikon E800 epifluorescent microscope or a Nikon E600 bright field microscope (Nikon, Melville, NY, USA), each linked to a digital camera (Gryphax Jenoptik, Jena, Germany) interfaced with imaging software (Bioquant Osteo 2022, Bioquant Image Analysis, Nashville, TN, USA) on a Windows 11 PC. The analysis of trichrome-staining-detected collagen deposition in muscles and nerves was performed using previously described methods [15]. The data for collagen staining in forepaw muscles were separated by regional location of the muscle quantified: in/around thenar, hypothenar, and deep intrinsic muscles in forepaws. This data were analyzed separately, as well as using a summed average.

### 4.7. Immunohistochemistry for αSMA and Collagen Type I

Additionally, sections containing muscles were immunostained in batched sets by the same individual for alpha smooth muscle actin (αSMA, A2547, Sigma-Aldrich, Burlington, MA, USA, dilution 1:500) or collagen type I (C2456, Sigma-Aldrich, 1:500 dilution in PBS), using previously described methods [15]. DAPI was used as a nuclear stain (62246, Thermo Fisher Scientific, Rockford, IL, USA; diluted 1: 2000 with PBS for 15 min) before coverslipping with 80% glycerol in PBS. Additional sections containing median nerve branches were similarly immunostained for collagen type 1.

### 4.8. Histomorphometry of Entheseal Damage

Potential degenerative changes were examined in carpal and metacarpal entheses in the trichrome stained sections, and scored using described methods [97]. Entheses were graded binarily (1 indicating present, 0 as not) on six domains: (1) tidemark changes, (2) underlying bone remodeling, (3) fissuring, (4) void space, (5) vascular invasion, and (6) attachment site holes. The presence of any of these changes is an index of degeneration [97]. The data for the enthesis scoring were separated by regional location of the entheses: lateral versus medial in the forepaw and wrist. These data were analyzed separately, as well as using a summed average. 

### 4.9. Western Blot Analyses and Gelatin Zymography

Flash-frozen muscle as well as the bone and entheseal complexes were stored at −80 °C until use, at which time they were thawed on ice and homogenized, separately, as previously described [15]. Briefly, tissues were homogenized in 100 mL of sterile, ice-cold, phosphate-buffered saline (PBS) containing proteinase inhibitors (1 Pierce™ Protease Inhibitor Tablet, EDTA-free, A32965, Pierce Biotechnology, Inc., ThermoFisher Scientific, Rockland, IL, USA) with 50 mL of sterile PBS without calcium and magnesium (21-040-CM, Mediatech, Inc., Corning, Manassas, VA, USA). After centrifugation, supernatants were collected and frozen at −80 °C until assayed. Protein levels in each sample were determined using a bicinchoninic acid (BCA) protein assay kit.

Electrophoresis and Western blot analysis of muscle supernatants was performed for analysis of β-catenin, phosphorylated-β-catenin, CCN1, CCN3, and MMP-1 protein expression levels, using described methods [15], with methodological differences for each protein assessed as described below in detail. The iBright™ Prestained Protein Ladder was used (Thermo Fisher Scientific). Gels were immunoblotted onto nitrocellulose membranes. Membranes were Ponceau S stained, then destained before blocking for 1 h in 5% bovine serum albumin in TBST and then incubated with primary antibodies overnight at 4 °C. All primary antibodies were diluted in Can Get Signal buffer (NKB 101T, Toyobo, Japan). After washing, blots were incubated with appropriate secondary antibodies diluted as described below in the Can Get Signal buffer (incubated 2 h at room temperature). Images were taken using a LI-COR System (Lincoln, NE, USA). Densitometric analysis was performed using Image J (ImageJ.org, National Institute of Health, Bethesda, MA, USA). Ponceau S stained densitometric data for each individual lane were used to normalize the density of target protein signal in each individual lane after antibody staining, as previously described [15].

The antibody used to detect β-catenin (B-9) was from Santa Cruz (sc-376841, made in mouse, Santa Cruz Biotechnology Inc., Dallas, TX, USA), diluted 1:200. The antibody used to detect phosphorylated-β-catenin (BC-22) was also from Santa Cruz (sc-57535, made in mouse), diluted 1:200. Both were detected using a donkey anti-mouse secondary antibody tagged with IRDye 800 CW (929-80020, LI-COR), diluted 1:20,000. Gels for both were 4–12% Tris-Glycine Wedge-Well gels (XP041220BOX, Novex, Thermo Fisher Scientific) loaded with 20 µg protein per lane, with homogenate supernatants previously boiled for 5 min at 100 °C, and prepared with beta-mercaptoethanol (BME).

The CCN1 antibody used was from R&D (AF4055, R&D, Minneapolis, MN, USA), diluted 1:1000. The secondary antibody was a rabbit anti-sheep Alexa Flour 680 (SA5-10058, Invitrogen, Carlsbad, CA, USA), diluted 1:15,000. Gels for CCN1 were 10% SDS Tris-Glycine Gel (XP00105BOX, Thermo Fisher Scientific) loaded with 40 µg of protein per lane. Samples were not boiled, yet prepared with BME. The secondary antibody used to detect CCN1 was a rabbit anti-sheep Alexa Flour 680 (SA5-10058, Invitrogen), diluted 1:15,000. Beta-tubulin (BTSR) DyLight 800 (MA5-16308-D800, Invitrogen, made in mouse), diluted 1:20,000, was used to reprobe the CCN1 probed membranes.

The antibody used to detect CCN3 was from R&D (AF1976, mouse NOV/CCN3), diluted 1:1000. The secondary antibody was a donkey anti-goat 800 (926-32214, LI-COR), diluted 1:40,000. The recombinant CCN3 protein was from R&D (1976-NV). Similar to CCN1, 10% SDS Tris-Glycine gels were used for CCN3 detection with unboiled samples prepared with BME.

The antibody used to detect MMP-1 was from Abbiotec (250750, Abbiotec, Escondido, CA, USA), diluted 1:300. A 10% SDS Tris-Glycine Gel was made for this purpose, with samples boiled and prepared with BME, and lanes loaded with 40 µg of protein each. The Intercept (PBS) Blocking Buffer (927-70001, LI-COR) was used for the MMP-1 gels.

Gelatin zymography was performed as previously described [15]. Purified recombinant rat MMP9 and MMP2 were used as positive controls (5427-MM-010 and 924-MP-010, respectively; R&D). SDS-PAGE gels (8%) were prepared with 0.05% gelatin according to the directions of the SureCast Handcast System (HC1000, Invitrogen, Life Technologies, Carlsbad, CA, USA). Lanes were loaded with 20 µg of protein per lane. PageRuler NIR Ladder (26635, Pierce Biotechnology, Inc., Thermo Fisher Scientific) was utilized for the gelatin zymograms.

The specificity of anti-β-catenin is shown in Figure 2 by including a membrane lane stained with the appropriate secondary antibody only and in Figure 8 with the inclusion of primary fibroblast lysates grown as previously described for primary tenocytes [19]. The specificity of the anti-CCN1 has been previously described [98], and here by including a membrane lane stained with the appropriate secondary antibody only. The specificity of anti-CCN3 is shown in Figure 4 by incubating the primary antibodies with the recombinant protein to CCN3 for two hours before incubating the membranes, and in Figure 8 by including a membrane lane stained with the secondary antibody only. MMP2 and 9 bands were verified by the inclusion of recombinant MMP 2 and 9 on the gelatin gels.

### 4.10. Statistical Analyses

This is the fourth study in a series examining behavioral outcomes and changes in different tissues from the same group of animals [33,34,37]. Details of the power analyses have been previously provided [37]. We chose the most conservative sample size for collagen staining and behavioral changes needed to detect differences with an alpha level of 0.05 and 80% power. This power analysis indicated the estimated sample size needed was 5 per group. Therefore, at least five per group and per assay were utilized in each of these studies. 

GraphPad, Prism 10, for Macs was used for the statistical analyses. Muscle and nerve collagen staining was assayed using Kruskal–Wallis nonparametric tests, followed by Dunn’s post hoc multiple comparison tests, as were the percent changes in behavior outcomes. Other data with only group as a factor (αSMA+ fibroblasts, CCN1, and phosphorylated-β-catenin Western blot data) were analyzed using one-way ANOVA followed by post hoc multiple comparison tests (Tukey or Fisher’s least significant difference). For analyses with multiple factors (muscle collagen staining data after muscle group subdivision, entheses features, Western blots in which multiple bands were detected), repeated-measures mixed-effect models were used, followed by Tukey’s multiple comparison post hoc tests. Pearson’s r or Spearman’s rank correlational analyses, as appropriate for the data, were used to compare the % change in grip strength or % change in forepaw mechanical sensitivity to tissue findings. An alpha level of less than 0.05 was used as significant, after adjustment for multiple comparisons. Data are expressed as mean ± SEM.

## 5. Conclusions

CCN2 is known to modulate several signaling pathways and to interact with many protein partners. Our past data suggest that blocking CCN2 signaling lowers not only its own levels, but also TGF-β1 and FGF2 levels (each pro-fibrogenic), αSMA and phosphorylated ERK immunopositive myofibroblast numbers in muscles, and collagen types 1 and 3 levels, and reduces established collagen deposition in forelimb neuromuscular tissues [33,34]. We have also shown that blocking CCN2 signaling increases osteoblast activity in bones of mature rats, a response that restored and even enhanced trabecular bone volume [37]. We extend those findings here to show that blocking CCN2 signaling ameliorated established forepaw neuromuscular fibrosis and entheseal damage following work-related musculoskeletal injury. The anti-CCN2 treatment also lowered levels of phosphorylated-β-catenin and elevated levels of CCN3 in muscles and entheses. These combined responses resulted in restoration of neuromuscular and enthesis structure, and, importantly, improved tissue function to control levels. These results support the feasibility of conducting clinical trials using an anti-CCN2 in human patients with overuse-induced fibrosis causing significant declines in muscle function.

## Figures and Tables

**Figure 1 ijms-24-13866-f001:**
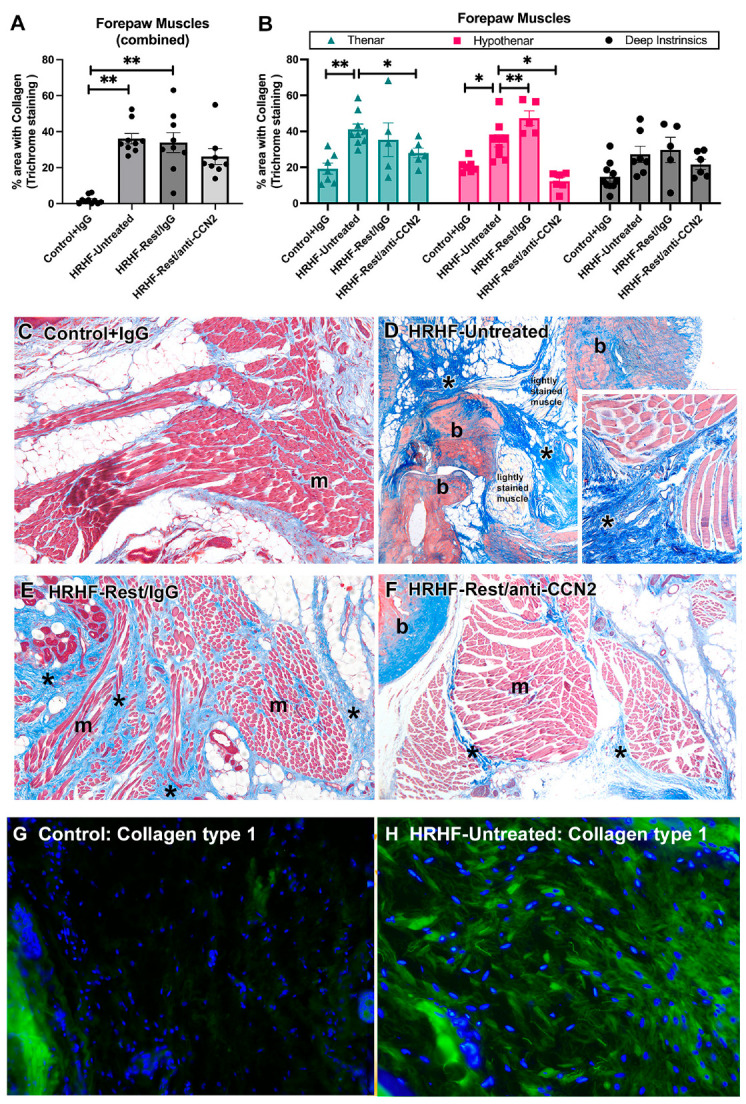
Fibrosis within and around forepaw muscles. (**A**) Percent area with collagen staining in all forepaw muscles examined combined. A Kruskal–Wallis test was performed, followed by Dunn’s post hoc tests. (**B**) Muscle data divided into thenar, hypothenar, and deep intrinsic (interossei) muscles. A repeated-measures mixed-model effect analysis was performed, followed by Tukey’s multiple comparison tests. Mean ± SEM is shown. *: *p* < 0.05 and **: *p* < 0.01, compared between groups as shown. (**C**–**F**) Representative images of hypothenar muscle region from each group in which sections were stained with Masson’s Trichrome staining (collagen is blue), with b = bone, m = muscle; * indicates areas of enhanced collagen deposition. (**G**,**H**) Representative images of extracellular matrix region between muscles of Control and HRHF-Untreated rats immunostained for Collagen type 1 (green fluorescence) with DAPI (blue) nuclear counterstaining. Magnification for panels (**C**–**F**) is 100×; inset in (**D**) is 200× magnification; panels (**G**, **H**) are 200×.

**Figure 2 ijms-24-13866-f002:**
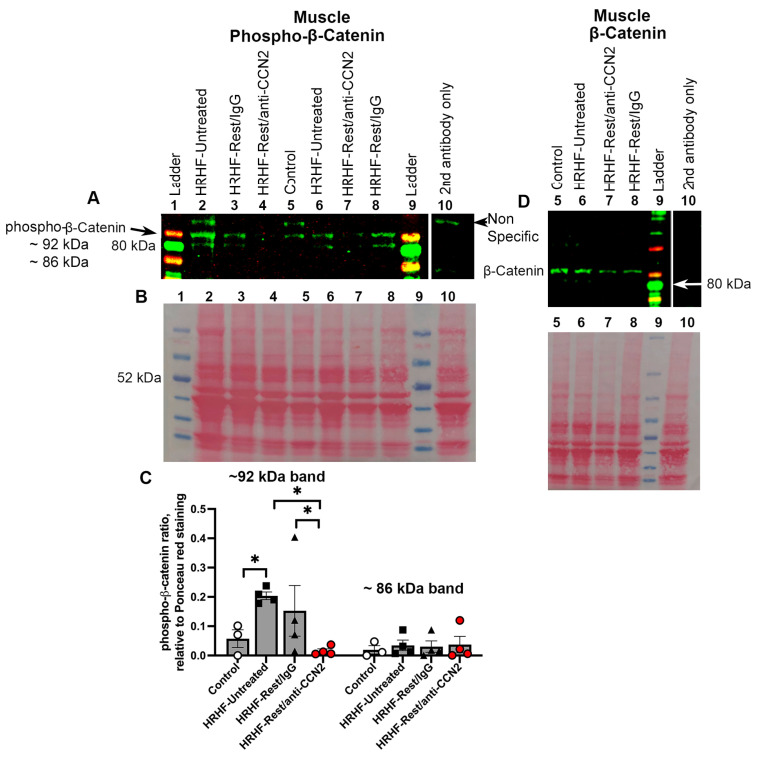
β-catenin protein levels in forelimb muscles. (**A**) Representative Western blot probed with anti-phosphorylated β-catenin. Lanes loaded as shown. Note: samples loaded in lanes 7 and 8 are reversed in order from lanes 3 and 4. Two specific bands, ~92 and ~86 kDa, were detected. (**B**) Matching Ponceau S red stained membrane, used to show protein loading levels. (**C**) Quantification of the phosphorylated β-catenin ~92 and ~86 kDa bands with expression shown as a ratio to the Ponceau S red staining in the matching lane. A repeated-measures mixed-effect model analysis was performed (with the two bands as the repeats) followed by Fisher’s LSD post hoc tests. Mean ± SEM is shown. *: *p* = 0.05, compared between groups as shown. (**D**) Representative blot probed with antibody to nonactive β-catenin in the same muscles, with the gel loaded as shown. Similar results to blot shown in (**D**) was observed in two replicate blots.

**Figure 3 ijms-24-13866-f003:**
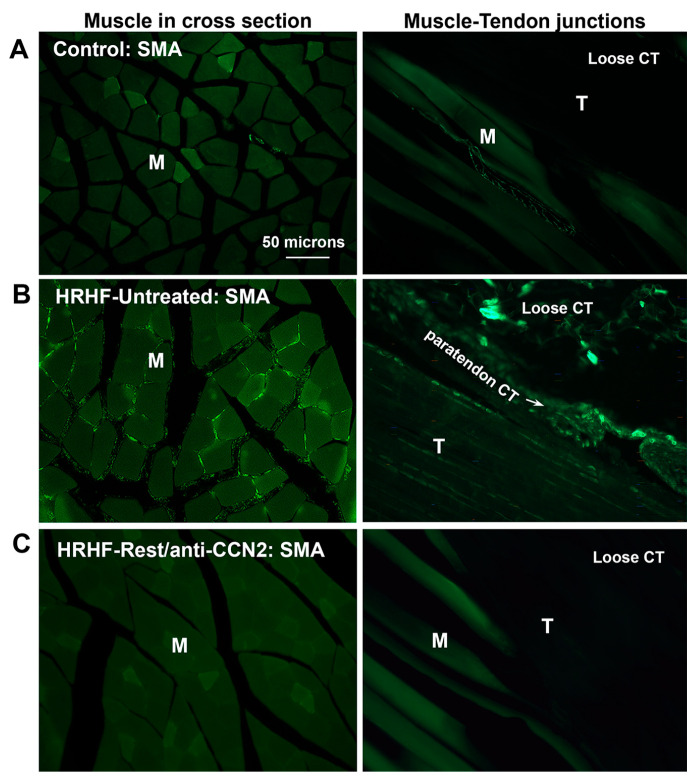
Representative images of alpha smooth muscle actin (αSMA) immunopositive cells (green immunofluorescence) in forelimb muscles and adjacent tissues. (**A**) Control. (**B**) HRHF-Untreated. (**C**) HRHF-Rest/anti-CCN2. The left column of images shows cross-sectionally cut muscles. The right column of images shows longitudinally cut sections of muscle–tendon junctions. M = muscle, T = tendon; Scale bar in panel (**A**) right applies to all other images.

**Figure 4 ijms-24-13866-f004:**
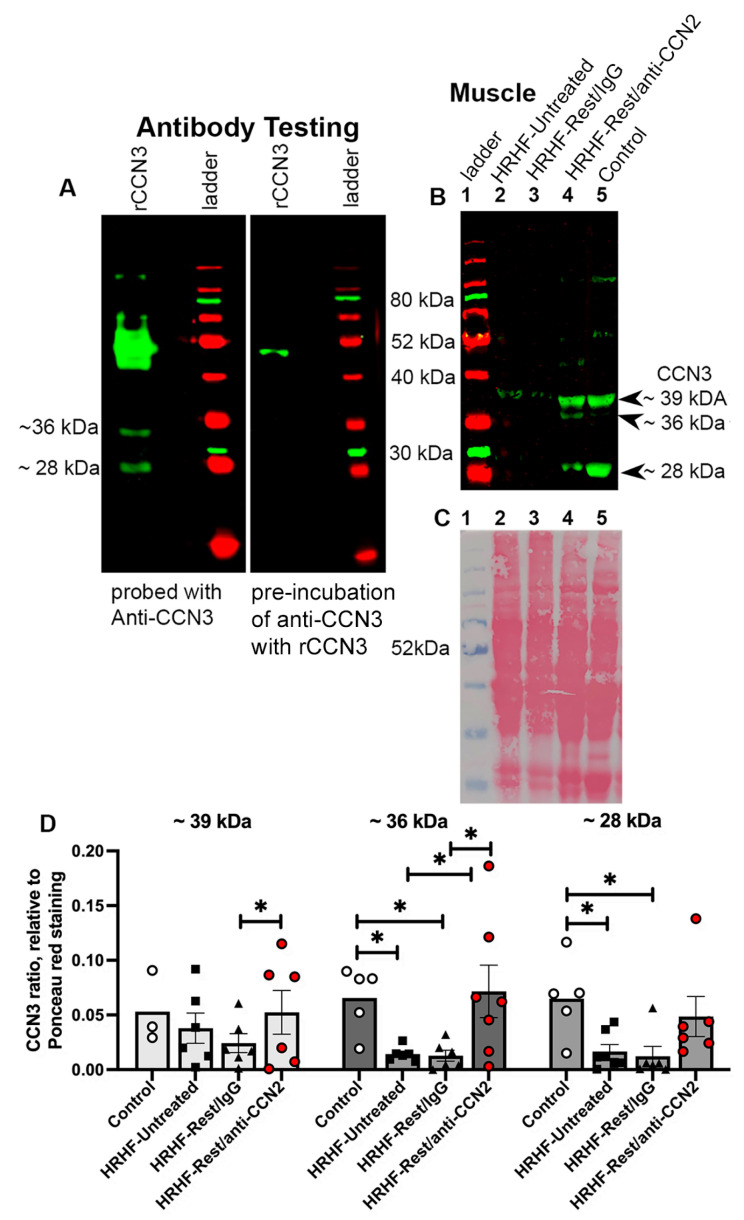
Anti-CCN3 probed Western blots. (**A**) Blots loaded with recombinant CCN3 (rCCN3) and probed with the CCN3 antibody with or without pre-absorption with the rCCN3. (**B**) Western blots of muscle lysates probed with the same anti-CCN3 antibody. Lanes loaded as shown. (**C**) Ponceau S staining of the same membrane shown in (**B**). (**D**) Densitometry results in which CCN3 bands were compared to the total protein loaded per lane, determined from Ponceau-S stained membranes. Mean ± SEM is shown. Results were assayed using repeated measures mixed models, followed by Fisher’s LSD post hoc tests *: *p* = 0.05, compared between groups as shown.

**Figure 5 ijms-24-13866-f005:**
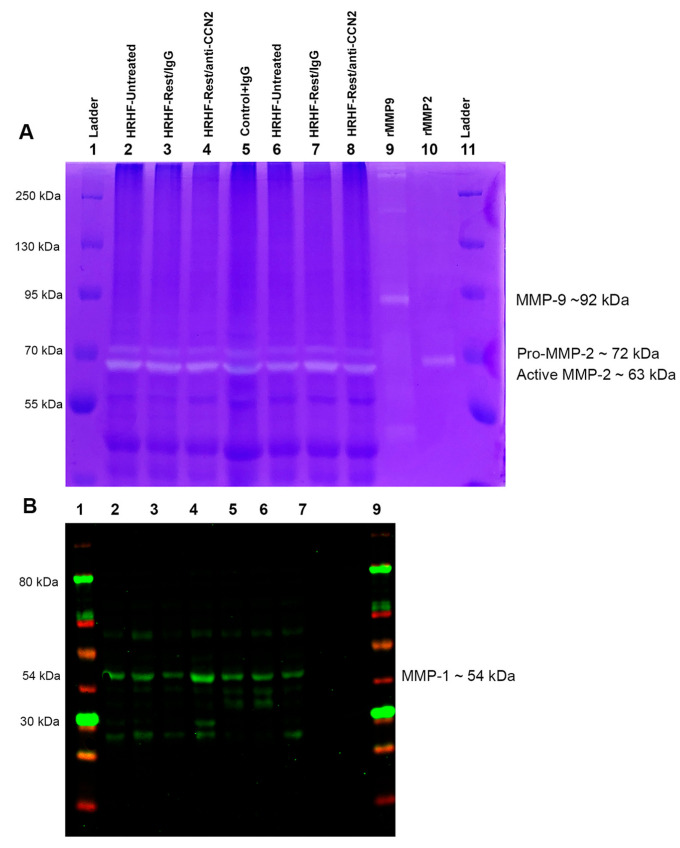
Zymography and Western blot analysis of muscle lysates for MMP-1, -2, and -9. (**A**) Gelatin zymography was used to assess MMP-2 and -9 expression. Lanes were loaded as shown. Recombinant MMP-2 and -9 were used to confirm bands detected (no MMP-9 was observed). (**B**) Western blot of muscle lysates probed with a specific antibody to MMP-1. Lanes loaded as described in (**A**) (lane 8 was not loaded). A band at the known molecular weight for MMP-1 (~54 kDa) was detected in each sample. Replicate zymograms and Western blots showed similar results.

**Figure 6 ijms-24-13866-f006:**
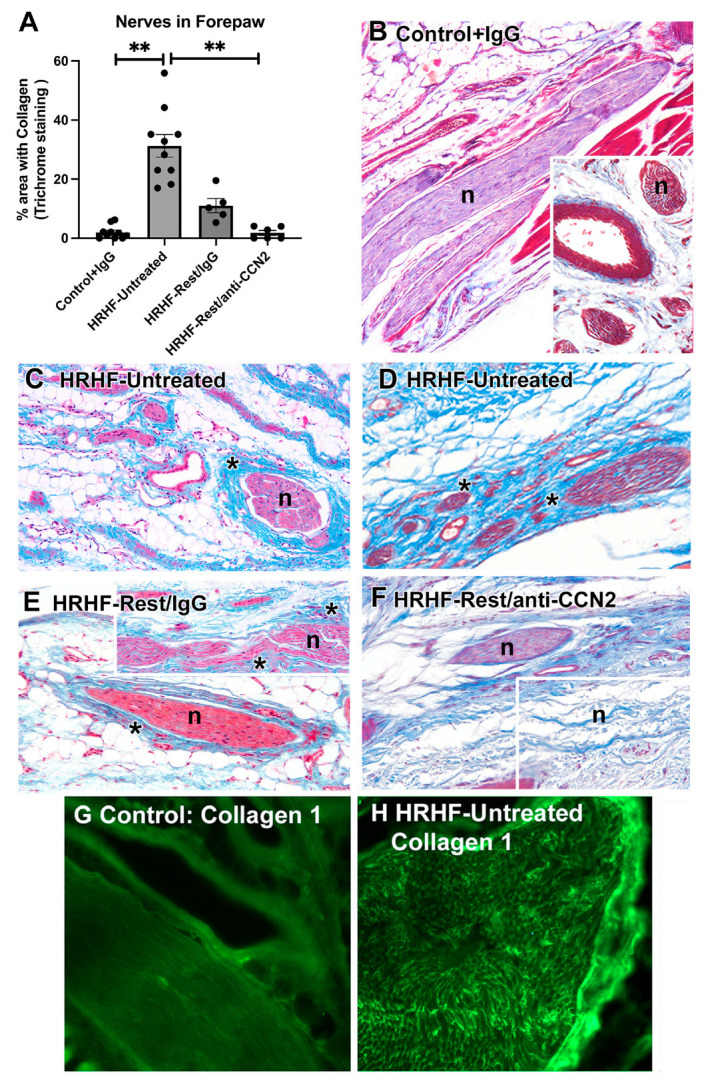
Fibrosis around forepaw nerves. (**A**) Percent area with collagen staining around nerve branches in the forepaw region. Kruskal–Wallis test performed, followed by Dunn’s post hoc comparison tests. Mean ± SEM is shown. ** *p* < 0.01, compared between groups as shown. (**B**–**F**) Representative nerve branches from each group stained with Masson’s Trichrome in which collagen is blue. Insets show additional examples. Nerve profiles are shown as longitudinal sections in panels (**B**,**D**,**E**) and inset, and (**F**); and as cross-sectional profiles in panels (**B**,**F**) insets, (**C**,**D**). Asterisks indicate representative areas of collagen deposition; *n* = nerve. (**G**,**H**) Nerves immunostained for collagen type I. Magnification for each image in panels (**B**–**H**) is 200×.

**Figure 7 ijms-24-13866-f007:**
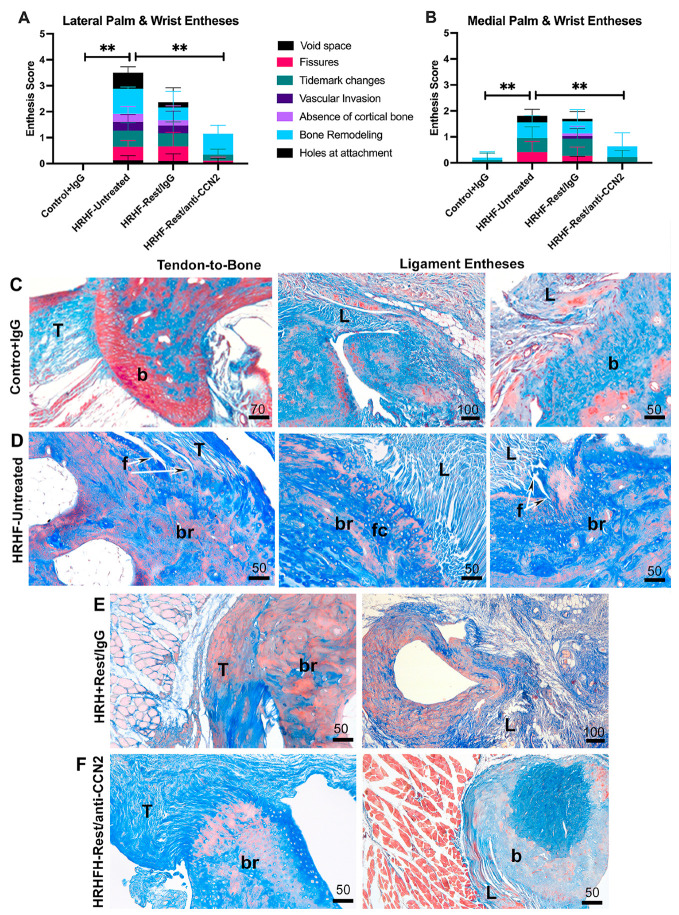
Forepaw and wrist entheses changes, subdivided into laterally versus medially located entheses. (**A**) Scores for laterally located entheses features shown. (**B**) Scores for medially located entheses features shown. A Kruskal–Wallis test performed for the sums, followed by Dunn’s post hoc tests with main effects shown. Mean ± SEM is shown. ** *p* < 0.01, compared between groups as shown. Enthesis features are listed in the middle of the panel. (**C**–**F**) Representative images of entheses from each group. Left images show tendon-to-bone entheses. Middle images in panels (**C**,**D**), and all right images show ligament (bone-to-bone) entheses. Abbreviations: b = bone, br = site of bone remodeling, f = fissures, fc = fibrocartilage islands within undecalcified bone remodeling areas, L = ligament, m = muscle, T = tendon. Scale bars indicated are in μm.

**Figure 8 ijms-24-13866-f008:**
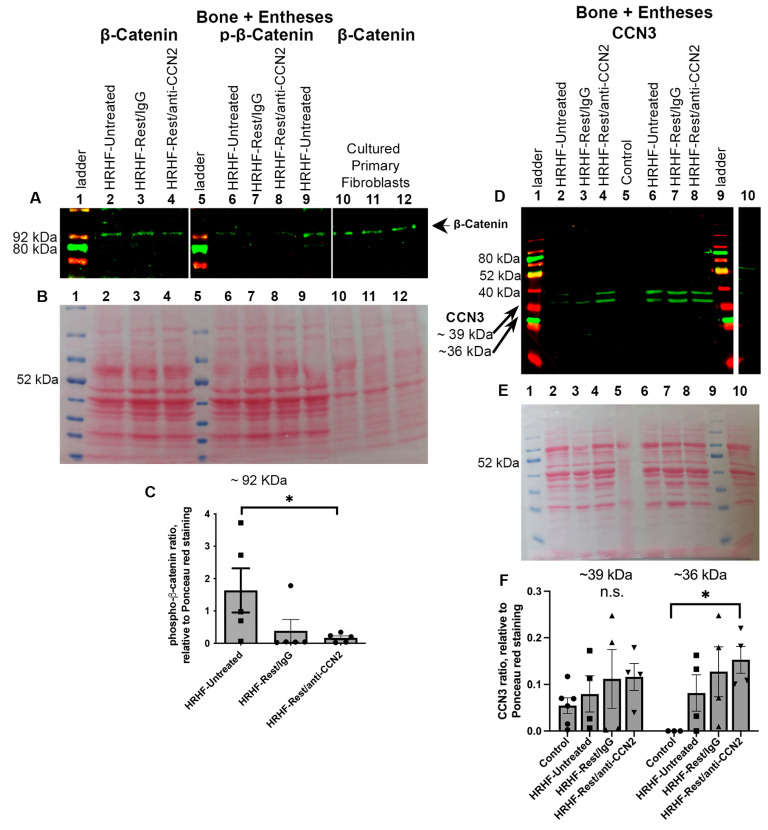
Western blots of distal forelimb/forepaw bone and associated entheses. (**A**) Blots probed β-catenin and phosphorylated (p) β-catenin. Lanes loaded as shown. The fibroblasts were grown in media containing 8% serum. (**B**) Matching Ponceau red stained membrane. (**C**) Quantification of the ~92 kDa p-β-catenin band as a ratio to the Ponceau staining. (**D**) Immunoblots probed with anti-CCN3. Lanes loaded as shown. One-way ANOVA followed by Fisher’s LSD post hoc tests. (**E**) Matching Ponceau stained membrane. (**F**) Quantification of the detected CCN3 bands as a ratio to the Ponceau red staining, used to normalize for protein loading. Mean ± SEM is shown. Repeated measures mixed model analysis followed by Fisher’s LSD post hoc tests. *: *p* < 0.05, between groups as shown; n.s. = not significant.

**Figure 9 ijms-24-13866-f009:**
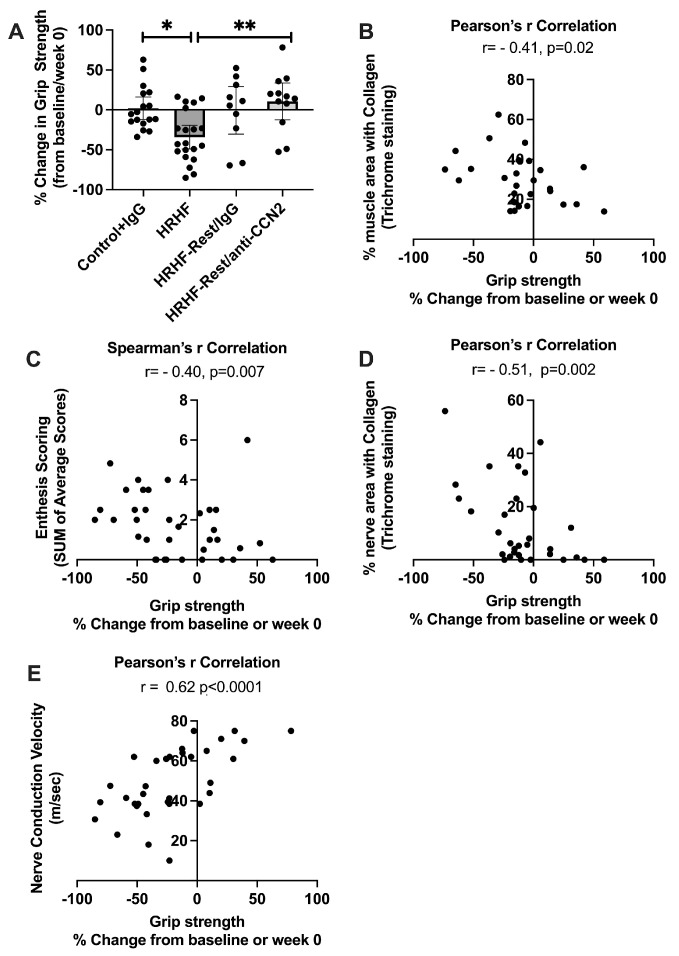
Percent change in grip strength and correlations with tissue or functional nerve changes. (**A**) Percent change in reflexive grip strength from baseline results, by the final week, showing a decline in grip strength in HRHF rats. A Kruskal–Wallis was performed, followed by Dunn’s post hoc tests with * *p* < 0.05 and ** *p* < 0.01, compared between groups as shown. (**B**–**D**) Correlations between % change in grip strength and % area forepaw muscle collagen staining, total enthesis score, and % area forepaw nerve collagen staining, respectively. (**E**) Correlation between % change in grip strength and median nerve conduction velocity in the final week.

**Figure 10 ijms-24-13866-f010:**
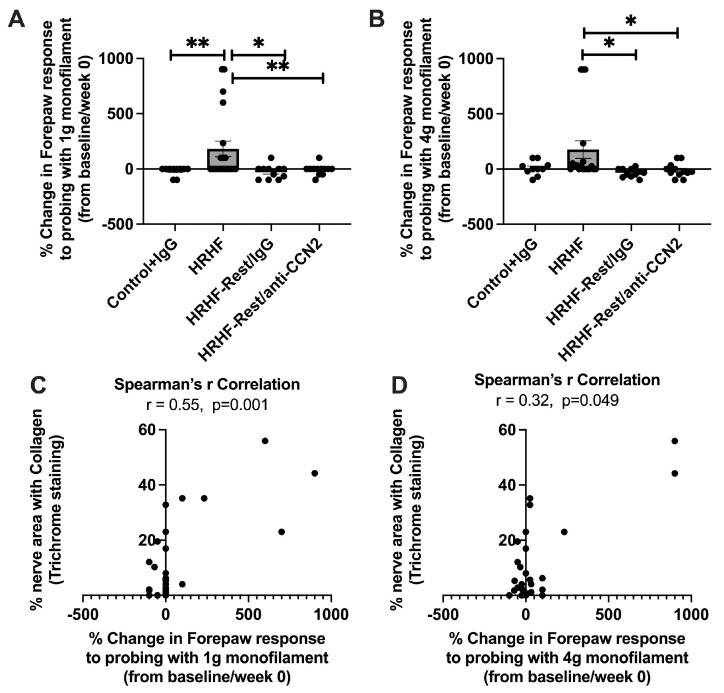
Percent change in mechanical sensitivity and correlations with forepaw tissue or functional nerve changes. (**A**,**B**) Percent change in forepaw response to probing with 1 g (0.98 cN) and 4 g (3.92 cN) sized monofilaments in the final week, respectively, from baseline results. An increase in percent change in this forepaw response is indicative of an increasing forepaw sensitivity to this non-noxious touch stimuli (i.e., hypersensitivity). A Kruskal–Wallis test was performed, followed by Dunn’s post hoc tests; * *p* < 0.05 and ** *p* < 0.01, compared between groups as shown. (**C**,**D**) Correlation between % area with nerve collagen staining and % change in the forepaw response to probing with 1 g and 4 g monofilament, respectively.

**Table 1 ijms-24-13866-t001:** Quantification of alpha smooth muscle actin immunopositive (αSMA+) cells in muscle.

Tissue	Control (*n* = 7)	HRHF-Untreated (*n* = 6)	HRHF-Rest/IgG (*n* = 5)	HRHF-Rest/Anti-CCN2 (*n* = 6)	ANOVA Results
Flexor Muscles	14.83 ± 2.91	290 ± 40.22 ^aa^	178 ± 57.65 ^aa,bb^	42.45 ± 12.35 ^b,cc^	*p* < 0.0001

Means ± SEM shown; ^aa^: *p* < 0.01, compared to Controls; ^b^: *p* < 0.05 and ^bb^: *p* < 0.01, compared to HRHF-Untreated; ^cc^: *p* < 0.01, compared to HRHF-Rest/IgG.

## Data Availability

Data is contained within the article and in the Appendix A.

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
