# Peer review of "Blocking CCN2 Reduces Established Palmar Neuromuscular Fibrosis and Improves Function Following Repetitive Overuse Injury"

_ijms, 2023, doi:10.3390/ijms241813866_

Round 1

Reviewer 1 Report

The paper Lambi et al., is one of a series of HRHF repeats in the forepaw, resulting in extensive fibrosis and connective tissue accumulation.

Here, they study the effect on muscle fibrosis (not new), neural fibrosis, and etheseal damage. The effect of resting +/- anti-CCN2 was evaluated.

The results are interesting but rather preliminary. The fibrotic response only is sustained by Masson’s trichrome staining (collagen type 1 likely). It would be very interesting and necessary to evaluate the level of other skeletal muscular extracellular components such as fibronectin, periostin, etc, as well as neural analyses and bone fibrosis. Particularly after rest, to know if fibrosis decreases in general or if some components are more resilient in the tissue.

The paper is rather preliminary and requires further experiments.

Also, it would be essential to know if fibroblasts are decreasing after resting and/or after anti-CCN2 injections. 

The behavioral assays are highly appreciated.

There are other papers using anti-CCN2 in different models of skeletal muscle fibrosis and should be cited.

Summarizing, this paper requires further experiments.

Author Response

Reviewer 1:

Comments and Suggestions for Authors

The paper Lambi et al., is one of a series of HRHF repeats in the forepaw, resulting in extensive fibrosis and connective tissue accumulation.

Here, they study the effect on muscle fibrosis (not new), neural fibrosis, and etheseal damage. The effect of resting +/- anti-CCN2 was evaluated.

Point 1: The results are interesting but rather preliminary. The fibrotic response only is sustained by Masson’s trichrome staining (collagen type 1 likely). It would be very interesting and necessary to evaluate the level of other skeletal muscular extracellular components such as fibronectin, periostin, etc, as well as neural analyses and bone fibrosis. Particularly after rest, to know if fibrosis decreases in general or if some components are more resilient in the tissue.

Response:

We have expanded the original objective of this paper and have now nearly doubled its size. We have now included western blot analyses of beta-catenin, CCN1 and CCN3 for muscle, as well as collagen type 1 immuohistochemistry for muscle and nerve to show that collagen type 1 is increased in control vs HRHF-Untreated. We also added a table to the appendix listing ELISA levels of TGF-beta 1, CCN2, collagen types 1 and 3, to further address this concern. We also added cell counts for alpha SMA in muscle as well as MMP-1, -2, and -9 results (reasons given in text for latter).

The Reviewer must have missed the extensive neural analysis presented in the original Figure 2, Figure 4D and E, and Figure 5. These are now in Figures 6, 9, and 10.

We do not see bone fibrosis in this model. Bone fibrosis is an uncommon type of bone marrow cancer, called myelofibrosis. This is an overuse injury model. Instead, to address degenerative changes in the bone/entheses, we added results of western blot analyses of beta-catenin, CCN1, and CCN3 of these tissues

Point 2: The paper is rather preliminary and requires further experiments.

Response: We performed further experiments as requested. We have nearly doubled the size of this manuscript now as we have included western blot analyses of beta-catenin, CCN1 and CCN3 for muscle, as well as collagen type 1 immuohistochemistry for muscle and nerve to show that the collagen type 1 is increased in control vs HRHF-Untreated. We also added a table to the appendix listing ELISA levels of TGF-beta 1, CCN2, collagen types 1 and 3, to further address this concern. We also added cell counts for alpha SMA in muscle as well as MMP-1, -2, and -9 results (reasons given in text for latter).

Also, it would be essential to know if fibroblasts are decreasing after resting and/or after anti-CCN2 injections. 

Response:  Numbers of SMA+ fibroblastic cells in muscle have been added.

The behavioral assays are highly appreciated.

Response:  Thank you.

There are other papers using anti-CCN2 in different models of skeletal muscle fibrosis and should be cited.

Response: We have added these papers. We apologize for this oversight.

Summarizing, this paper requires further experiments.

We performed further experiments as requested. We have nearly doubled the size of this manuscript now as we have now included western blot analyses of beta-catenin, CCN1 and CCN3 for muscle, as well as collagen type 1 immuohistochemistry for muscle and nerve to show that collagen type 1 is increased in control vs HRHF-Untreated rats. We also added a table to the appendix listing ELISA levels of TGF-beta 1, CCN2, collagen types 1 and 3, to further address this concern. We also added cell counts for alpha SMA in muscle as well as MMP-1, -2, and -9 results (reasons given in text for latter).

Reviewer 2 Report

The manuscript of “Blocking CCN2 Reduces Established Palmar Neuromuscular Fibrosis and Improves Function following Repetitive Overuse Injury” by A.G. Lambi” and co-authors aims to examine whether rest, with or without concurrent anti-CCN2 treatment, would reduce degenerative changes in the forelimb tissues and improve function, grip strength and forepaw mechanical sensitivity. Using histological analysis, the authors demonstrated that blocking CCN2 with an anti-CCN2 monoclonal antibody reduced forepaw neuromuscular fibrosis and entheseal damage following work-related musculoskeletal injury. The manuscript may contribute to the development of new potential therapeutic strategies, but the underlying molecular mechanisms of action of human anti-CCN2 monoclonal antibody in a rat model of overuse injury have not been investigated. As a result, the manuscript is rather descriptive.

Comments:

1.      In fact, the authors used only one method, namely Masson’s Trichrome staining, to reveal neuromuscular fibrosis and entheseal damage in a rat model of overuse injury. The underlying molecular mechanisms of development and progression of repetitive overuse injury, as well as the mechanism of action of human anti-CCN2 monoclonal antibody in rats with repetitive overuse injury have not been investigated. The discussion of the results obtained is speculative and it does not clarify the role of the matricellular protein connective tissue growth factor (CTGF/CCN2) in the progression of forelimb neuromuscular fibrosis and functional declines in a rat model of overuse injury.

2.      Lines 438-440: The authors need to explain why the intraperitoneal route (but not intramuscular injections) of human anti-CCN2 monoclonal antibody administration to rats was chosen. Lines 406-415: It is also not clear why the number of animals in the treatment groups was reduced by half, compared with that in the control groups.

Author Response

Reviewer 2

Comments and Suggestions for Authors

The manuscript of “Blocking CCN2 Reduces Established Palmar Neuromuscular Fibrosis and Improves Function following Repetitive Overuse Injury” by A.G. Lambi” and co-authors aims to examine whether rest, with or without concurrent anti-CCN2 treatment, would reduce degenerative changes in the forelimb tissues and improve function, grip strength and forepaw mechanical sensitivity. Using histological analysis, the authors demonstrated that blocking CCN2 with an anti-CCN2 monoclonal antibody reduced forepaw neuromuscular fibrosis and entheseal damage following work-related musculoskeletal injury. The manuscript may contribute to the development of new potential therapeutic strategies, but the underlying molecular mechanisms of action of human anti-CCN2 monoclonal antibody in a rat model of overuse injury have not been investigated. As a result, the manuscript is rather descriptive.

Comments:

  1. In fact, the authors used only one method, namely Masson’s Trichrome staining, to reveal neuromuscular fibrosis and entheseal damage in a rat model of overuse injury. The underlying molecular mechanisms of development and progression of repetitive overuse injury, as well as the mechanism of action of human anti-CCN2 monoclonal antibody in rats with repetitive overuse injury have not been investigated. The discussion of the results obtained is speculative and it does not clarify the role of the matricellular protein connective tissue growth factor (CTGF/CCN2) in the progression of forelimb neuromuscular fibrosis and functional declines in a rat model of overuse injury.

Response:  This current paper is not focused on understanding the underlying molecular mechanisms of development and progression of repetitive overuse injury. We apologize for being unclear in the introduction.

We have now expanded on this point in the introduction. Our past studies examining underlying molecular mechanisms of development and progression of repetitive overuse injury began in 2003. We have cited only key studies here for succinctness. We also clearly delineated the involvement of CCN2 in the progression of forelimb neuromuscular fibrosis and functional declines in a rat model of overuse injury. This was published in a prevention study (Barbe et al, 2019) in which the anti-CCN2 was used early while the fibrosis was developing. Those findings were summarized, and that paper cited. For focus, we did not add our many studies examining the effectiveness (or not) of ibuprofen, NK1RA, treadmill running, or manual therapy, as preventives in the development and progression of repetitive overuse injury.

Instead, in this study, we are examining the reversal of established fibrosis and degenerative changes induced by the long term performance of an intensive overuse task. That has been clarified now throughout. We expanded on our past reversal findings as well in the introduction, focusing primarily on the ones using the anti-CCN2 (as we have also used rest, anti-NK1RA, and manual therapy as reversal treatments in prior studies from our lab).  

We have also expanded our analysis of histopathologic and molecular changes following additional experiments. These include immunohistochemistry of type 1 collagen in muscle and nerve, and alpha SMA positive cells in muscle. We also analyzed the levels in muscle of multiple proteins involved in fibrosis using western blot, zymograms, and ELISA (as discussed in the text). Proteins studied include beta-catenin (including active, phosphorylated beta-catenin), CCN2 protein family members CCN1 and CCN3, TGF-beta 1, CCN2, collagen types 1 and 3, and MMP-1, -2, and -9.

The one original paragraph on these points in the first submission have been expanded in the introduction into three paragraphs. Please see lines 55-110 in which we cover prevention, the involvement of CCN2, and then reversal experiments. We also re-organized and expanded the Discussion, and have include more of the results of others in the Discussion.

  1. Lines 438-440: The authors need to explain why the intraperitoneal route (but not intramuscular injections) of human anti-CCN2 monoclonal antibody administration to rats was chosen. Lines 406-415: It is also not clear why the number of animals in the treatment groups was reduced by half, compared with that in the control groups.

Response:  First, the company recommends an intraperitoneal route for animals and systemic route for humans. This was important to us, since we have previously shown that the fibrosis is widespread in many tissue types (muscle, nerve, tendon) in this animal model and in humans with upper extremity repetitive overuse injuries. The widespread nature in both muscles and nerve are also shown in this submission. Thus, localized intramuscular injections would not be sufficient as a treatment. A paragraph has been added to the discussion on this point.

Please see the paragraph beginning on line 632:

“To date, antibodies to CCN2 are administered systemically for fibrotic disorders of a number of types in humans and in vivo animal models [15, 26, 30-32, 35, 78-81]. This is an advantage for the treatment of overuse injuries that can effect a variety of different tissues (tendon, nerve, muscle, and bone) and multiple anatomical sites dependent on the work task [82-84], sometimes simultaneously in the same patient [7, 82, 83]. In our rat model of overuse injury, repeated and persistent inflammation and subsequent fibrogenic processes in forepaw and forelimb nerves, muscles, tendons, and associated connective tissues [15, 16], while degenerative changes have been detected in tendon and bone [30, 35, 85]. These changes are observed in all anatomical regions involved in performing the task (forepaws, as shown here, as well as forelimbs, shoulder, cervical and thoracic) [12, 86]. We postulate that the systemic administration of a drug is key to recovery of function in these types of disorders.”

This is the fourth of a series, as described; therefore matches the past designs and their power analyses. The power analysis was provided again here. Please see the paragraph beginning with line 847:

            “This is the fourth study in a series examining behavioral outcomes and changes in different tissues from the same group of animals [31, 32, 35]. Details of the power analyses have been previously provided [35]. We chose the most conservative sample size for collagen staining and behavioral changes needed to detect differences with an alpha level of 0.05 and 80% power. This power analysis indicated the estimated sample size needed was 5 per group. Therefore, at least five per group and per assay were utilized in each of these studies.

Round 2

Reviewer 1 Report

My comments and suggestions have been adequately addressed.

Reviewer 2 Report

The manuscript has been substantially revised and improved. I have no further concerns.